

# High throughput microRNAs sequencing profile of serum exosomes in women with and without polycystic ovarian syndrome

Feng Zhang[1,2,*], Su-Ping Li[3,*], Tao Zhang[1,2], Bin Yu[1,2], Juan Zhang[1,2], Hai-Gang Ding[1,2], Fei-Jun Ye[4], Hua Yuan[1,2], Ying-Ying Ma[1,2], Hai-Tao Pan[1,2] and Yao He[1,2]

[1] Shaoxing Maternity and Child Health Care Hospital, Shaoxing, China
[2] Obstetrics and Gynecology Hospital of Shaoxing University, Shaoxing, China
[3] Jiaxing University Affiliated Women and Children's Hospital, Jiaxing, China
[4] Zhoushan Maternity and Child Health Care Hospital, Zhoushan, China
[*] These authors contributed equally to this work.

## ABSTRACT

**Background**. Polycystic ovary syndrome (PCOS) is the most common type of endocrine disorder, affecting 5–11% of women of reproductive age worldwide. microRNAs (miRNAs) stably exist in circulating blood encapsulated in extracellular vesicles such as exosomes; therefore, serum miRNAs have the potential to serve as novel PCOS biomarkers.

**Methods**. To identify miRNA biomarkers that are associated with PCOS, we performed a comprehensive sequence-based characterization of the PCOS serum miRNA landscape. The serum exosomes were successfully isolated and characterized in a variety of ways. Next, sequence-based analysis was performed on serum exosomes to screen the differentially expressed miRNAs in women with and without PCOS.

**Results**. The sequence data revealed that the levels of 54 miRNAs significantly differed between PCOS patients and normal controls. The levels of these miRNAs were detected by RT-qPCR. The results show that hsa-miR-1299, hsa-miR-6818-5p hsa-miR-192-5p, and hsa-miR-145-5p are significantly differentially expressed in PCOS patients serum exosomes and identify these microRNAs as potential biomarkers for PCOS. Furthermore, Gene Ontology (GO) analyses and KEGG pathway analyses of the miRNA targets further allowed to explore the potential implication of the miRNAs in PCOS.

**Conclusion**. Our findings suggest that serum exosomal miRNAs serve important roles in PCOS and may be used as novel molecular biomarkers for clinical diagnosis.

Corresponding authors
Hai-Tao Pan,
panhaitao2007@163.com
Yao He, sxfbheyao@163.com

## INTRODUCTION

Polycystic ovary syndrome (PCOS) is the most common infertility disorder in reproductive-aged women. Metabolic issues like inflammation, increased coagulability, visceral obesity, insulin resistance and androgen excess are considered as key features in PCOS (*Chen et al.,*

*2019*). PCOS is a multifactorial disorder that affects 5–11% of women of reproductive age worldwide (*Chen et al., 2019*). Exosomes are membrane-bound vesicles that are released into body fluids such as serum, plasma, urine and saliva (*Kalluri & LeBleu, 2020*). In particular, plasma and serum exosome microRNAs (miRNAs) profiling have been shown to have a potential in the diagnosis of different diseases (*LeBleu & Kalluri, 2020*).

Exosomes are a class of extracellular vesicles (EVs) that are secreted by most cell types of the body able to transport biologically active cargo between cells, including protein, lipids, mRNA and miRNA (*Kalluri & LeBleu, 2020*). microRNAs (miRNAs) are a class of endogenous and highly conserved non-coding small RNA, with the length from 18 to 24 nucleotides. It has been previously reported that the small RNA sequences contained in human follicular fluid (HFF) exosomes may play a key role in the mechanism that drives PCOS pathogenesis, and thereby can act as molecular biomarkers for PCOS diagnosis in the future (*Hu et al., 2020*). Mesenchymal stem cells derived exosomal miR-323-3p promotes proliferation and inhibit apoptosis of cumulus cells in polycystic ovary syndrome (PCOS) (*Zhao et al., 2019a*). PCOS serum-derived exosomal miR-27a-5p stimulates endometrial cancer cells migration and invasion, indicating that serum exosomal miR-27a-5p may play a role in EC development in PCOS patients (*Che et al., 2020*). circRNAs profile of follicle fluid exosomes has also been sequenced in polycystic ovary syndrome patients (*Wang et al., 2019*). In addition, S100-A9 protein in exosomes derived from follicular fluid promotes inflammation via activation of NF-kappaB pathway in polycystic ovary syndrome (*Li et al., 2020*). Although other studies have reported alterations in miRNA in human follicular fluid (HFF) exosomes and mesenchymal stem cells exosomes (*Zhao et al., 2019a*; *Che et al., 2020*; *Wang et al., 2019*; *Li et al., 2020*), there is still a need for further research in promising therapeutic and prognostic serum biomarkers in PCOS.

Based on previous studies, it is likely that serum exosomes and their contents, particularly miRNAs, may serve as potential biomarkers for PCOS. Therefore, this study aimed to investigate and compare the expression of these miRNAs in serum exosomes of women with and without polycystic ovary syndrome (PCOS). The findings of this study may provide a new biomarker and therapeutic target that facilitates early diagnosis and treatment for PCOS.

## MATERIALS & METHODS

### Study participants

The present study was approved by the Ethics Committee of Shaoxing Maternity and Child Health Care Hospital (Ethical Application Ref: 2018024). All participants signed an appropriate informed consent form. Serum samples from 4 healthy female donors were collected as normal control, while serum obtained from PCOS according to the Rotterdam standard served as the experimental group ($n = 4$). Their clinical characteristics are outlined in Table 1.

### Exosomes purification and characterization

To isolate exosomes, all samples were filtered through 0.22 μM filters. Exosomes were isolated from the serum sample using ExoQuick Exosome Precipitation Solution (System

**Table 1  Clinical information of PCOS group and normal group.**

| Parameter | Reference range | PCOS group (*n* = 4) | Normal group (*n* = 4) | *P*-value |
|---|---|---|---|---|
| Age (years) | | $28.5 \pm 1.73$ | $27.50 \pm 1.29$ | 0.390 |
| BMI (kg/m2) | | $20.85 \pm 1.35$ | $20.45 \pm 0.62$ | 0.610 |
| FSH (IU/L) | 3.3–7.9 | $7.92 \pm 0.74$ | $5.56 \pm 0.32$ | 0.001 |
| LH (IU/L) | 2.0–12.0 | $16.38 \pm 2.00$ | $8.55 \pm 1.85$ | 0.001 |
| AMH (ng/ml) | 3.31–7.98 | $11.30 \pm 1.22$ | $6.21 \pm 1.31$ | 0.001 |
| PRL (ng/ml) | 3.5–24.2 | $16.23 \pm 4.72$ | $15.33 \pm 3.18$ | 0.742 |
| E2 (pmol/L) | 18–195 | $64.11 \pm 13.24$ | $53.83 \pm 10.60$ | 0.271 |
| Progesterone (ng/mL) | 0.2–1.2 | $0.72 \pm 0.15$ | $0.52 \pm 0.28$ | 0.259 |
| Testosterone (ng/mL) | 0.15–0.51 | $0.71 \pm 0.20$ | $0.38 \pm 0.09$ | 0.022 |

Notes.
AMH, antimullerian hormone;  BMI, body mass index;  FSH, follicle-stimulating hormone;  LH, follicle-stimulating hormone;  PCOS, polycystic ovary syndrome;  PRL, prolactin.

Biosciences, CA, USA). Exosomes were characterized by electron microscopy (FEI Quanta 250; Thermo Fisher Scientific, Eindhoven, the Netherlands), nano particle tracking analysis (NTA) (ZetaView PMX 110; Particle Metrix, Meerbusch, Germany), and Western blot analysis (for CD9, HSP70).

## Exosomes miRNA sequencing and sequence analysis

Samples were collected from 4 PCOS patients and 4 non-PCOS volunteers. The kit and protocol used were the Illumina TruSeq RNA Sample Preparation kit v2 (catalog # RS-122-2001) and the TruSeq RNA Sample Preparation v2 Guide (part # 15026495). The libraries were sequenced on an Illumina HiSeq X Ten platform. Sequencing was performed by paired - end sequencing ($2 \times 150$ bp) on Illumina HiSeqX to a minimal depth of $30 \times$ base coverage. Raw sequence alignment and variant calling were carried out using Illumina CASAVA 1.8 software. To quantify microRNAs, back-spliced junction reads were scaled to RPKM mapped reads.

miRWalk (http://mirwalk.uni-hd.de) and miRBase 22.0 (http://www.mirbase.org) were used to analyze the miRNA sequences. The FastQC software (https://www.bioinformatics.babraham.ac.uk/projects/fastqc/) was used to make sure that the sequencing data were high quality. The heatmap was plotted based on the log2(fold change), using Heatmap Illustrator software (Heml 1.0). In addition, gene ontology (GO) and pathway analyses were performed to identify miRNA-related genes, pathways and GO terms based on sequencing data sets. GO analysis was performed using DAVID (http://david.abcc.ncifcrf.gov/). The RNAhybrid and miRanda database were used to predict potential target genes of all the differentially expressed miRNAs. Set-up parameters were as follows: RNAhybird energy <25; miRand a energy <20 and score>−150.

## Western blotting analysis

Western blotting was performed as we previously described (*Pan et al., 2018*), four exosomes of human serum were prepared from control groups and four samples from PCOS groups, separately. Samples were separated in a 10% SDS gel. The separated samples

**Table 2  The microRNA primers used in the study.**

| microRNA | primers |
|---|---|
| hsa-miR-1299 | 5′-UUCUGGAAUUCUGUGUGAGGGA-3′ |
| hsa-miR-6818-5p | 5′-UUGUGUGAGUACAGAGAGCAUC-3′ |
| hsa-miR-6818-5p | 5′-CUGACCUAUGAAUUGACAGCC-3′ |
| hsa-miR-145-5p | 5′-GUCCAGUUUUCCCAGGAAUCCCU-3′ |

were transferred to a nitrocellulose transfer membrane (Bio-Rad, Hercules, CA, USA). After incubating for 1 h with blocking buffer, the membrane was incubated overnight at 4 °C with primary antibodies against Hsp70 (Beyotime AF1156, Shanghai, China, 1:1,000), CD9 (Beyotime AF1192, Shanghai, China, 1:1,000) , beta Tubulin (Beyotime AF1216 , Shanghai, China, 1:1,000) and Calnexin (Beyotime AC018, Shanghai, China, 1:1,000) at 4 °C overnight. After three washes with −TBST,pH 7.4, each membrane was incubated with the appropriate secondary antibody (1: 1000) at room temperature for 1 h. After additional three washes, protein intensities were visualized with the enhanced ECL detection system (Sage Creation, Beijing, China). Each experiment was repeated three times.

### RT-qPCR analysis
RT-qPCR was performed as we previously described (*Pan et al., 2018*). Total miRNAs from the serum exosomes were extracted using the miRcute miRNA isolation kit (TIANGEN, Beijing, China). Real-time PCR was performed for validation using miRcute miRNA qPCR Detection Kit (SYBR Green) (TIANGEN, Beijing, China) after polyadenylation and reverse transcription (miRcute miRNA First-Strand cDNA Synthesis Kit, TIANGEN, China). All miRNA primers were purchased from Tiangen (Beijing, China). The relative microRNA levels were normalized to U6 expression for each sample. The microRNA primers used in the study are presented in Table 2. The reactions were performed with a StepOne Plus Real-Time PCR System (Applied Biosystems) and StepOne software v2.1. The PCR reaction included a fast start step of 15 min at 95 °C followed by 45 cycles of amplification where each cycle consisted of denaturation at 94 °C for 20 s and annealing at 60 °C for 34 s. Melting curve analysis was performed to verify the identity and specificity of the PCR products. Analyses of gene expression was performed by the $2^{-\Delta\Delta Ct}$ method. Each experiment was repeated three times.

### Statistical analysis
Data are displayed as the means ± standard deviations (SD). The statistical significance of the results from three independent assays was evaluated by Student's $t$-test. $P < 0.05$ was considered to indicate statistically significant differences.

# RESULTS
### Clinical information of PCOS group and normal group
Exosomes were purified from the normal controls ($^{Ct}$) and PCOS participants' ($t$) serum. PCOS patients had a mean age of 28.1 ($n = 41.72$) years and healthy controls had a mean age of 26.8 ($n = 41.27$) years. There were no significant differences of body mass index

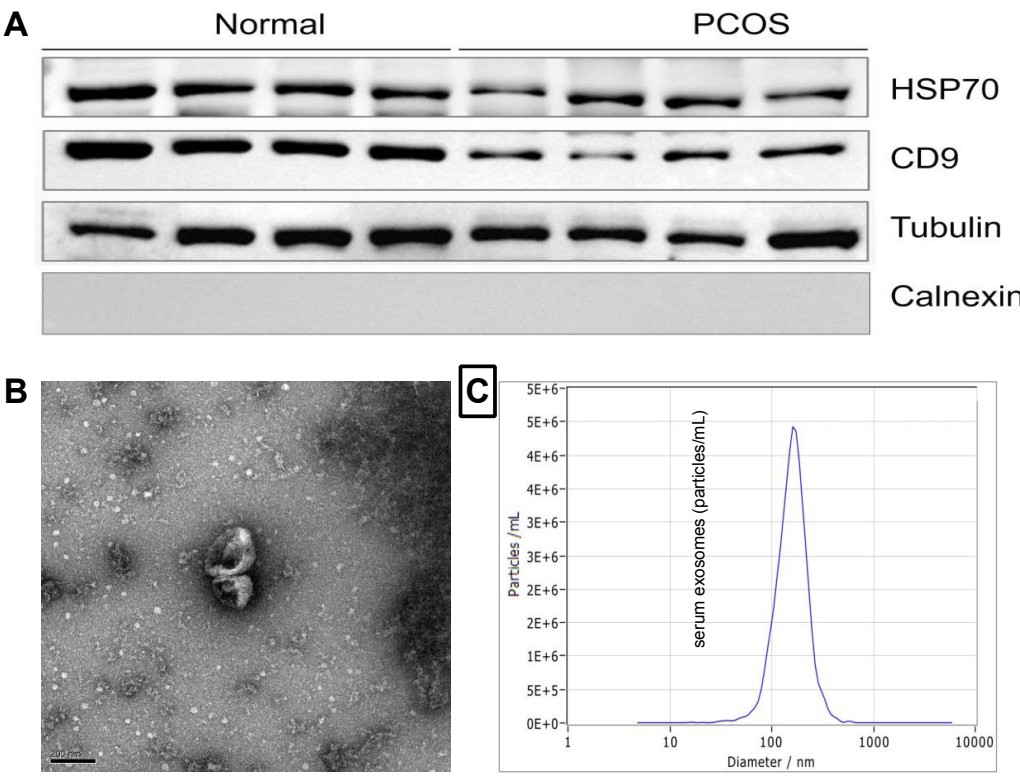

**Figure 1** **Characterization of exosomes derived from human serum.** (A) Western blot analysis of the surface markers of exosomes (Hsp70 and CD9). (B) Serum-derived exosomes visualized by electron microscopy (scale bar, 200 nm). (C) The representative nanoparticle tracking analysis (NTA) profile of exosomes from human serum.

(BMI) between the patients and the matched controls. Participant clinical characteristics are shown in Table 1.

## Isolation and characterization of serum-derived EXOs

The characteristics and properties of isolated exosomes are described in Fig. 1. Western blot analysis validated the expression of the known exosomal biomarkers CD9 and HSP70 (Fig. 1A). The size and morphology feature of exosomes were examined by transmission electron microscopy (scale bar, 200 nm) (Fig. 1B). NTA analysis showed that exosomes isolated from serum had a diameter size range of ∼154.7 ± 9 nm (Fig. 1C), consistent with the known size of exosomes.

## Analysis of miRNAs in serum EXOs

In the present study, a total of 1475 miRNAs were identified in serum exosomes from both PCOS and the controls (Table S1). The original sequencing data were uploaded to the NCBI SRA database (https://www.ncbi.nlm.nih.gov/sra) with the accession number PRJNA646028. Statistical analysis of the exosomal microRNAs sequencing results revealed that the levels of 54 mature miRNAs in serum exosomes from the PCOS group differed significantly from those in control group ($P < 0.05$, $\log_2|FC| > 1.2$) (Table S2). As shown
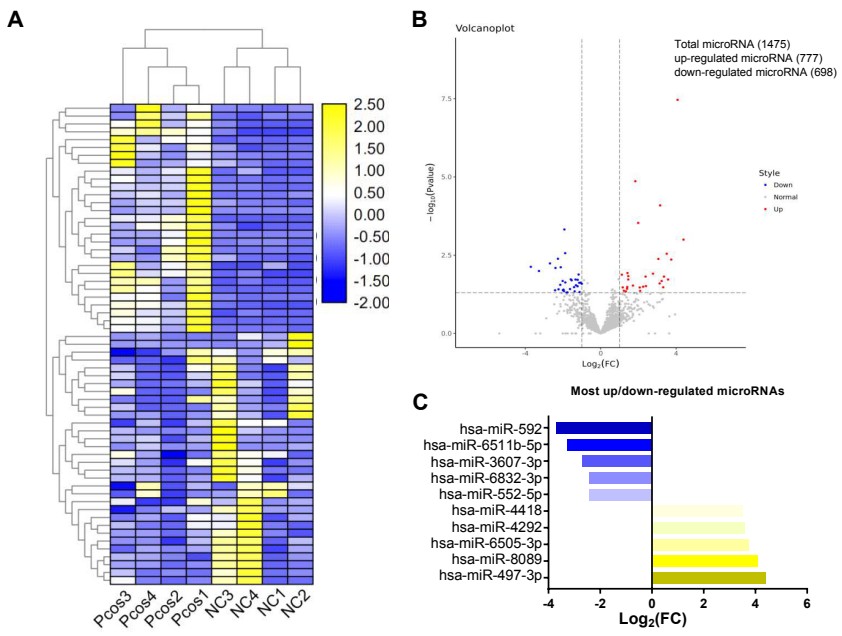

**Figure 2** **Cluster analysis of differentially expressed exosomal miRNAs isolated from serum from PCOS patients and normal controls.** (A) Heat map with hierarchical clustering (HCL) of normalized microRNAs abundance reveals the 54 differentially expressed microRNAs from PCOS ($p < 0.05$) and normal control ($\geq$). (B) Volcano map showing the distribution of differential microRNAs according to their $n = 4$ values and fold-changes. Candidates with $p < 0.05$ and $|\log2(\text{fold} - \text{change})| \geq 1$ are considered differential. (C) The top five most up-and downregulated microRNAs.

in Fig. 2, hierarchical clustering for the 54 differentially expressed miRNAs, including 27 upregulated miRNAs and 27 downregulated miRNA, is indicated by a heatmap and a volcano plot.

qPCR was performed for miRNAs with significant changes based on the miRNA sequencing data (Fig. 3). We confirmed upregulation of hsa-miR-1299, hsa-miR-6818-5p and downregulation of hsa-miR-192-5p, hsa-miR-145-5p in serum exosomes from women with and without PCOS. In addition, we confirmed that the FANCC is a miR-1299 target gene (Fig. S4).

## GO and pathway analysis of miRNA target genes

It was observed that 54 differentially expressed miRNAs can target 28,502 differentially expressed genes (Fig. S1, Table S3). Furthermore, Gene Oncology (GO) (Fig. S2) and Kyoto Encyclopedia of Genes and Genomes (KEGG) pathway analysis (Fig. 4) was performed to investigate the functions of their target genes. In addition, KEGG pathways enrichment revealed that the DEGs were mainly involved in the axon guidance, pathways in cancer and MAPK signaling pathway.

## DISCUSSION

Scrutinizing the role and differential expression of miRNAs in various tissues of PCOS patients has been the subject of increasing number of studies, which generally focused on
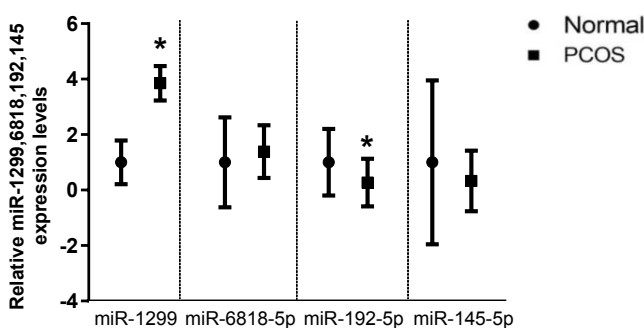

**Figure 3** **Validation of selected miRNAs by qRT-PCR.** $*p < 0.05$; miRNA: micro RNA; PCOS: polycystic ovary syndrome; qRT-PCR: quantitative reverse-transcription polymerase chain reaction.

miRNAs as potential biomarkers of the syndrome (*Jiang et al., 2015*). Multiple independent studies have reported that exosomes contain miRNA, and exosomes have been proposed to be treasure chests for biomarker applications (*Cheng et al., 2014*). Furthermore, miRNAs have recently emerged as ideal molecules that could be used as diagnostic and prognostic biomarkers in many diseases (*Zhang et al., 2019*). In this study, we performed a comprehensive sequence-based characterization of the PCOS serum exosomal miRNAs, the sequence data revealed that the levels of 54 miRNAs significantly differed between PCOS patients and normal controls. As shown in Fig. S3, the PCA graph has enabled a good discrimination between the samples. The PCA two-dimensional spatial distribution was consistent with that of the heat map analysis. Our findings in this study demonstrate that hsa-miR1299, hsa-miR-6818-5p hsa-miR-192-5p, and hsa-miR-145-5p are differentially expressed in PCOS patients serum exosomes and have the potential to be used as biomarkers in the diagnosis of PCOS.

Recently, protein profile (*Li et al., 2020*), small RNA sequences (*Hu et al., 2020*) and circRNAs profile (*Wang et al., 2019*) of follicle fluid exosomes have been identified in polycystic ovary syndrome patients (*Wang et al., 2019*). In addition, exosomal miR-323-3p and miR-27a-5p also have been proved may play a role in polycystic ovary syndrome (PCOS). Similarly to *Che et al. (2020)*, we identified potential exosomal miRNAs and their gene targets using sequencing and bioinformatics approaches. However, several of these miRNAs are potentially novel biomarkers for PCOS that have not been mentioned in previous reports. To confirm this, underlying epigenetic mechanisms should be further investigated.

Some studies focusing on alteration miRNAs have been published. miR-1299 has recently been reported as potential biological markers for rheumatic heart disease (RHD) development (*Li et al., 2015*). Gestational Diabetes Mellitus Regulatory Network Identifies hsa-miR-145-5p as Potential Biomarkers (*Zamanian Azodi et al., 2019*). hsa-miR-192-5p has the potential to be an early diagnostic marker for Venous thrombosis (VT) (*Rodriguez-Rius et al., 2020*). hsa-miR-4649-5p has the potential to be Amyotrophic Lateral Sclerosis (ALS) diagnosis biomarkers (*Takahashi et al., 2015*). miR-6782-5p exhibit the highest expression and are candidate circulating biomarkers for metastatic activity of prostate

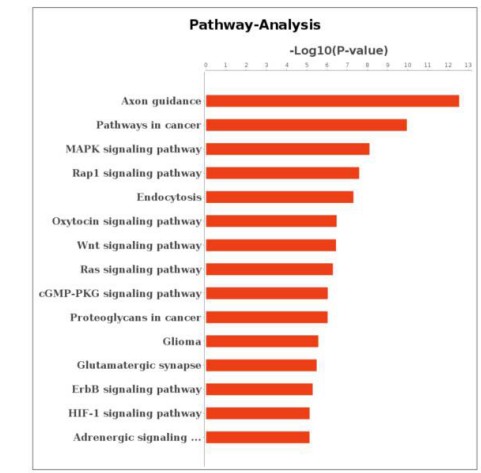

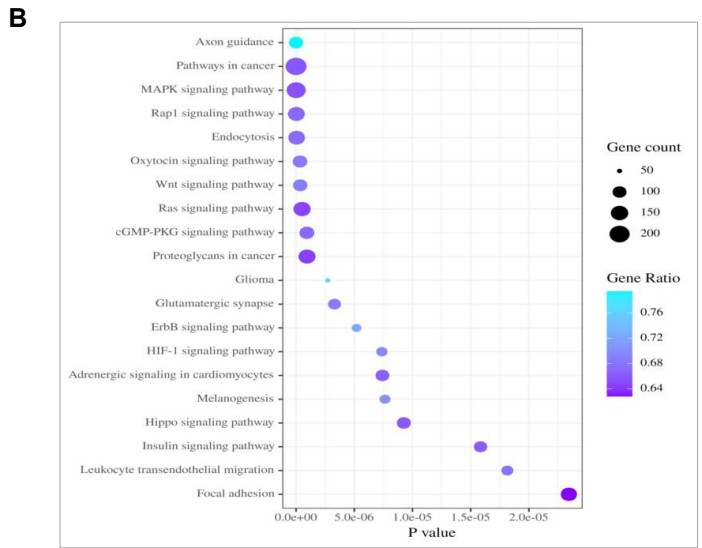

**Figure 4    KEGG pathway analysis for the predicted target genes of the differentially expressed miR-NAs.** (A) Coordinate axis X: $-\log_{10}$ ($P$-value); coordinate axis $Y$: Pathway-Term entry name. (B) Coordinate axis X: $-log_{10}$ ($P$-Value); coordinate .axis $Y$: Pathway-Term entry name.

cancer (*Fomicheva et al., 2017*). miR-4306 acts as a tumor suppressor in TNBC and is a potential therapeutic target for TNBC treatment (*Zhao et al., 2019b*). hsa-miR-31-5p plays an important role in HS formation by inhibiting FIH and regulating the HIF-1alpha pathway (*Wang et al., 2017*). hsa-miR-143-5p robustly down-regulated in Intracranial aneurysm (IA) (*Jiang et al., 2013*). Downregulation hsa-miR-214-3p may suppressed human epithelial ovarian cancer (EOC) development (*Wang et al., 2018*). hsa-mir125b-5p can be a prognostic and diagnostic biomarker for for Stage I Lung Adenocarcinoma (*Zeybek et al., 2019*). hsa-miR-135b-5p regulates the APC gene in both intestinal and diffuse subtypes of Gastric Cancer (GC) (*Magalhaes et al., 2018*). miR-5701 inhibits the proliferation of cervical cancer cells and the expression of its target gene THBS4 (*Prajapati*

*et al., 2018*). hsa-miR-3607-3p may represent as promising therapeutic targets against kidney fibrosis (*Yu et al., 2019*). hsa-miR-592 showed correlation with monosomy 3 tumors (*Wroblewska et al., 2020*). The significance of the different expressions of miRNA in various diseases remains to be determined. Therefore, exploring several key biomarkers for PCOS diagnosis and treatment will be of great significance.

miRNA target-based pathway enrichment analysis of PCOS revealed 284 enriched pathways that included MAPK signaling, Hippo signaling pathway and Insulin signaling pathway (Table S4); these pathways are known to be associated with PCOS. TRIB1 can activate the MAPK kinase pathway, which is induced in muscle cells in PCOS women (*Liu et al., 2013*). Hippo gene expression fingerprints could potentially be used to more accurately define patients with PCOS (*Maas et al., 2018*). Insulin resistance is considered as part of the pathogenesis of polycystic ovary syndrome (PCOS), and PCOS patients often show hyperinsulinemia (*Zhang et al., 2020*). The exact relationship between insulin resistance and Anti-Müllerian hormone (AMH) has not been fully elucidated. Recent research suggests that the diversity of AMH genotypes in the AMH signal pathway may be connected with the susceptibility and phenotype of PCOS with insulin resistance (*Sahmay et al., 2018*). Women with PCOS have high AMH; and accordingly AMH has been proposed as a marker of PCOS with high AFC (*Gupta et al., 2019*). AMH levels have the potential to be diagnostic and prognostic modalities in PCOS patients.

## CONCLUSION

Collectively, in our study, the exosomal microRNA expression profiles between PCOS patients and non-PCOS individuals were comparatively and comprehensively analyzed. hsa-miR-1299, hsa-miR-145-5p, hsa-miR-6818-5p and hsa-miR-192-5p may be potential biomarkers for diagnosis and treatment of PCOS. Nevertheless, identification of potential biomarkers of response was limited by the small sample size. Future research is needed to determine a better diagnostic criteria and ways to improve the diagnostic rate in early stages of PCOS.

### Funding

This work was supported by the National Natural Science Foundation of China (81701522 and 82071729), the Zhejiang Provincial Natural Science Foundation of China (LY19H040002), the Science Technology Department of Zhejiang Province, China (LGF21H040003, LGF21H040004, LGF19H040004, LGD20H040001, LGF18H180015) and the Health Commission of Zhejiang Province, China (2021KY375, 2021KY1154, 2021KY1157, 2020KY1001, 2018KY844, 2018KY845, 2018KY848, 2019KY229, 2019KY230, 2019RC296, 2019KY717); the Science Technology Department of Shaoxing, China (2020A13032, 2020A13034, 2020A13035, 2020A13037, 2018C30039, 2018C30042, 2018C30043, 2018C30044, 2018C30048). The funders had no role in study design, data collection and analysis, decision to publish, or preparation of the manuscript.

## Grant Disclosures

The following grant information was disclosed by the authors:

National Natural Science Foundation of China: 81701522, 82071729.

Zhejiang Provincial Natural Science Foundation of China: LY19H040002.

Science Technology Department of Zhejiang Province, China: LGF21H040003, LGF21H040004, LGF19H040004, LGD20H040001, LGF18H180015.

Health Commission of Zhejiang Province, China: 2021KY375, 2021KY1154, 2021KY1157, 2020KY1001, 2018KY844, 2018KY845, 2018KY848, 2019KY229, 2019KY230, 2019RC296, 2019KY717.

Science Technology Department of Shaoxing, China: 2020A13032, 2020A13034, 2020A13035, 2020A13037, 2018C30039, 2018C30042, 2018C30043, 2018C30044, 2018C30048.

## Competing Interests

The authors declare there are no competing interests.

## Author Contributions

- Feng Zhang, Su-Ping Li and Yao He conceived and designed the experiments, authored or reviewed drafts of the paper, and approved the final draft.
- Tao Zhang, Bin Yu and Ying-Ying Ma performed the experiments, prepared figures and/or tables, and approved the final draft.
- Juan Zhang and Hua Yuan analyzed the data, authored or reviewed drafts of the paper, and approved the final draft.
- Hai-Gang Ding and Fei-Jun Ye analyzed the data, prepared figures and/or tables, and approved the final draft.
- Hai-Tao Pan conceived and designed the experiments, prepared figures and/or tables, authored or reviewed drafts of the paper, and approved the final draft.
- conceived and designed the experiments, authored or reviewed drafts of the paper, and approved the final draft.

## Human Ethics

The following information was supplied relating to ethical approvals (i.e., approving body and any reference numbers):

The present study was approved by the Ethics Committee of Shaoxing Maternity and Child Health Care Hospital (Ethical Application Ref: 2018024).

## DNA Deposition

The following information was supplied regarding the deposition of DNA sequences:

The original sequencing data are available at the NCBI SRA database: PRJNA646028.

## Data Availability

Raw data in the Supplementary Files.

## Supplemental Information

Supplemental information for this article can be found online at http://dx.doi.org/10.7717/peerj.10998#supplemental-information.

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
