# Peer review of "High throughput microRNAs sequencing profile of serum exosomes in women with and without polycystic ovarian syndrome"

_PeerJ, doi:10.7717/peerj.10998_

## Round 0.1 · original submission · Major Revisions

Although the manuscript is conceptually innovative, the authors need to improve, in particular, the information about methodology, the analysis and interpretation of results, and their presentation.

Reviewer 1 ·

Basic reporting

Clear outline of the paper. Sound description of aims and objectives. Citations are relevant and recent.

Experimental design

Authors have employed a reasonable experimental design. The subject selection us clear. The manuscript provides detailed research methodologies although authors need to include some more details on statistical analysis.

Validity of the findings

This is an interesting and informative study; and the findings would contribute to the current knowledge in the field.

Additional comments

Manuscript submitted by Zhang et al., describes characterization of the miRNA profiles from serum exosomes isolated from patients with polycystic ovary syndrome (PCOS) and the healthy individuals. The study identifies 54 miRNAs that show statistically significant difference in the expression. Authors further confirm some of the identified candidates by biochemical assay (RT-qPCR) and show the evidence of their differential expression. The study is interesting and well designed. The role of miRNAs in exosomes as potential biomarkers is an exciting field in research investigation. The findings from this paper would contribute to the growing body of the evidences for the same. Here are my comments that would help authors to improve the quality of the manuscript and benefit general readership.

1) What are the downstream targets of miRNAs that authors identified to be differentially expressed by both sequencing AND RT-qPCR. Can authors provide any evidence of changes in expression of the target mRNAs (by reporter assays and/or Western blot)?
2) Can authors include a model depicting one their differentially expressed miRNAs in exosomes and the potential implications in disease development. It would be really helpful.
3) Authors need to mention how many times each experiment was repeated (e.g. Figure 1A: Western blot, Figure 3: RT-qPCR assay etc.).
4) For sequencing data: Authors need to comment on the data reproducibility across the four biological replicates (e.g. Pearson’s correlation and/or Principal component analysis).
5) Figure 1A: Did authors confirm the purity of exosomes by immunoblot analysis using any non-exosome biomarkers? Also, why is there a difference in the expression of both HSP70 and CD9 in PCOS samples as compared to control? Is this statistically significant and/or biologically relevant?
6) Line 157: Authors need to provide little more details of statistical analysis of the sequencing data.
7) Lines 164-166: These statements wrongly describe the results. Figure 3 shows significant changes (represented by *) in the expression of ONLY miR-1299 (up) and miR-192-5p (down) and NOT miR-6818-5p or miR-145-5p. The lines 190-192 in the discussion and the lines 222-223 in conclusions also wrongly describe these results. This needs to be corrected. Also, the legends (data-symbols) on Figure 3 need to be clearer. I assume circles represent ‘normal samples’ and squares represent ‘PCOS samples’. This needs to be clarified.
8) Figures 1B and 1C: Are these data from normal individuals or PCOS patients? It would be helpful to show the data from both the groups and comment on the general morphology of the exosomes, if any.
9) Line 154: Authors mention that the study identified 1475 miRNAs. What was the read-count cutoff used?
10) Figure 2B: What do the numbers in round brackets represent?
11) The fold-change criterion described on line 159 is not the same as described in Figure legends for 2B.
12) Line 77-78: Authors write: “Although some studies have reported alterations in miRNA expression in PCOS..” This needs relevant citation/s.
13) Figure 2C: X axis label is missing.
14) Legends for Supplementary Figures are missing.
15) There are a few grammatical and sentence construction errors that need to be corrected (e.g. line 71: ‘inhibits’ should read ‘inhibit’, line 74: “…follicle fluid exosomes was also has been..” should read “…follicle fluid exosomes has also been..”, line 78: “…expression in PCOS, there are still need to further study..” should read “…expression in PCOS, there is still need to further study…”).

Reviewer 2 ·

Basic reporting

The structure of the whole article is clear, but there are many formatting errors in the article. To sum up, the content is of interest and innovation, so this manuscript is recommended for publication before some issue modification.

Experimental design

This work started with PCOS and normal human serum exosomes, using miRNA sequencing technology to reveal the difference in miRNA expression and predict the novel molecular biomarkers for clinical diagnosis. It provides scientific help for clinical identification of PCOS.

Validity of the findings

All underlying data have been provided,but it needs to be added some more.

Additional comments

1. Line 159,P<0.05 should be in italics,There were many similar formatting errors in the article, please correct.

2. Line 159, |FC|>1.2 or log2|FC|>1.2?

3. Line 171, repeated punctuation.

4. Line173-174, these contents should be written in the method.

5. Figure2B legend, |log 2(fold‐change)| ≥1 or ≥1.2 ?

6. The legend in Figure 2B shows that the up-regulated microRNA is red and the number is 777, but it looks like there are only a few dozen. please explain.

7. Figure2C: Please indicate the unit of horizontal axis and vertical axis in detail.

8. I noticed that in your original data, the expression level of the hsa-miR-497-3p gene in the NC group is 0, but the log2FC value is 4.39. Please explain the calculation method to get the value of 4.39.

9. Please complete the legend in Figure 3

10. Figure 4A: What is the unit of the vertical axis? Target genes are enriched in many pathways. Why choose these pathways as the result display? What is the basis? Are these pathways significant? Please give the logical closeness between the pathway result and PCOS.

11. There are several errors in the legend in Figure 4A, please correct them.

12. Human_Participant_Consent_Form is an empty form, please complete it.

13. Because many miR expression levels are very low, I want to look at the quality control data of mirna sequencing to prove the accuracy of mirna sequencing.

14. Is it possible to add a Principal Component Analysis chart (PCA chart) based on the miRNA gene expression data of different samples to visually display the differences between and within groups.

Reviewer 3 ·

Basic reporting

Zhang et al have studied the differential regulation of miRNAs from serum exosomes in PCOS with matched controls using miRNA sequencing. They have characterized the exosomes and profiled the miRNAs and performed enrichment analysis.
The formatting of the authors' affiliation has to be checked and presented in conventional ways.
Conclusions section in the abstract is missing.
The language is clear and professional except for a few places where there are grammatical errors. The introduction is sufficient and discusses the relevant literature. The quality of the figures needs to be improved. Detailed comments on the figures are given in the general comments.

Experimental design

The scope of the work matches with that of the PeerJ and the research question is well defined.
However, the experimental methods are insufficiently explained with critical information missing making it difficult to assess the findings. The missing information has been explained in the general comments.

Validity of the findings

The study is novel and there is no known study reporting the exosomal miRNAs in PCOS compared to matched controls. The rationale and benefit to the literature have been clearly stated.
However, the data and methods which have been provided are inadequate to support the findings of the study.

Additional comments

For high throughput studies, it is always recommended to conduct the analysis with correcting for multiple testing and generally false discovery rate (FDR) corrected p-values is considered appropriate to define statistical significance. The study has used p-values to define differentially expressed miRNAs throughout the manuscript and have used those miRNAs for downstream analyses. While checking the raw data from this study (Suppl. Table S3), only 6 miRNAs passed multiple testing (FDR <0.25) and just 3 miRNAs passed more stringent criteria (FDR <0.05). The authors have chosen to proceed with the uncorrected p-values for enrichment analysis. Please justify this,
Methods
In methods under section 2.3, most of the critical information in methods are missing. No information has been presented on the library preparation, illumina instrument used for sequencing, depth of sequencing, sequence alignment, mapped reads, how differential expression of miRNAs was quantified and GO analysis was performed. All these critical information needs to be included in the methodology section.
Section 2.5
Details of PCR including the temperature and no of cycles, primers is missing.
The sentences in section 3.4 169-171 and 172-174 should ideally be placed in Methods rather than in results.
Results
Figure 1: The loading control is missing which is critical to ensure equal loading.
Figure 2: The information on how the heatmap was constructed is missing. Were the values normalized and scaled? Legend is missing. Also, please check the discrepancy in the volcano plot which mentions 777 up-regulated and 698 downregulated miRNAs while the results section lines 158-160 indicate that 54 miRNAs were differentially regulated.
Figure 3: Please check the missing legends
Figure 4: Please provide the tool used to perform KEGG pathway enrichment. In Figure 4A, the Y-axis is poorly defined. In Figure 4B, dot plot has been wrongly mentioned as scatter plot. In addition, a part of the legend, “Pathway enrichment is measured by Rich factor, P value and the number of genes enriched in this pathway. Rich factor refers to the ratio of the number of differentially enriched genes to the number of annotated genes in the Pathway. The larger the Rich factor, the greater the degree of enrichment” is identical to the legend from Hu et al 2020 – Figure 6. Please rephrase the legends.
Discussion
The observations from the studies can be put together in a better manner. There is a list of miRNA showing their links with other conditions. The possible links between the newly identified miRNAs and PCOS have to be discussed. The major 6 miRNAs and their targeting genes can be checked against the known RNA markers of PCOS. Also, it is important to discuss about the enriched pathways with respect to PCOS. The observation of high Anti-Müllerian hormone in PCOS has not been discussed with respect to the differentially expressed miRNAs.
The conclusive statements are difficult to understand.

Supplemental figures
The figures are missing legends.

Annotated reviews are not available for download in order to protect the identity of reviewers who chose to remain anonymous.

---

## Round 0.2 · Minor Revisions

Some issues raised by reviewer 1 and some clarifications on Fig. 4, as requested by reviewer 3, need to be better addressed.

Reviewer 1 ·

Basic reporting

No additional comments.

Experimental design

No additional comments.

Validity of the findings

No additional comments.

Additional comments

In the revised manuscript authors have addressed many of my comments but some of them still remain unresolved.
1) In response to my previous comment regarding miRNA downstream target gene expression analysis – authors provided western blot data for miR-1299 target gene FANCC. The data clearly show decreased levels of protein and convincingly demonstrate the effects of increased levels of miRNA-1299. But I did not find these data included in the revised manuscript. I think biochemical validation of the findings is a significant addition to the existing data and it strengthens the claims. Please include these data and add relevant text in pertinent sections of the manuscript.
2) My second comment has not been addressed satisfactorily as well. I was anticipating a more thorough and comprehensive summary of the findings including possible target mRNAs (and their possible biological role in disease). I agree with authors that there remains much need for further elucidation. But based on the available data, can authors include a speculative (and more comprehensive than they showed in the rebuttal letter) schematic model? Also, authors need to make sure to include the model in not only the the rebuttal letter but also the manuscript as well (may be as a supplementary figure; if not the main data).
3) Lines 81 – 84 [File: peerj-49119-Manuscript_(tracked_changes)]: Authors write – “Although other studies have reported alterations in miRNA in human follicular fluid (HFF) exosomes and mesenchymal stem cells exosomes, there is still a need for further research in promising therapeutic and prognostic serum biomarkers in PCOS”. I had commented in my earlier review that this needs relevant citation/s of the published literature. Authors claim (in their rebuttal letter) that they added the citations. But I do not see the relevant literature is cited here. Please do the needful.
4) I am still not convinced the way authors have defined “differential regulated microRNAs” and “significant differentially regulated miRNAs”. If I understand correctly, the ONLY difference between these two groups is – for the former, the fold-change cutoff is 2-fold while for the latter it is 2.3-fold. Why did authors employ this arbitrary cutoff to define latter set of miRNAs? It is surprising (but not practically impossible) to get the number restricted from 1475 to only 54 when the cutoff stringency is just marginally increased. Please clarify this discrepancy.
5) Upon reading the manuscript again, I realized that a statement on depositing the raw sequencing data (data availability) is missing. Have authors deposited their raw sequencing data to a publicly available repository? To ensure reproducibility, these data need to either be cited or deposited into an appropriate repository and a clear statement needs to be included in the manuscript.

Reviewer 2 ·

Basic reporting

The structure of the whole article is clear, many formatting errors in the article have been corrected. To sum up, the content is of interest and innovation and this manuscript is recommended for publication.

Experimental design

This work started with PCOS and normal human serum exosomes, using miRNA sequencing technology to reveal the difference in miRNA expression and predict the novel molecular biomarkers for clinical diagnosis. It provides scientific help for clinical identification of PCOS.

Validity of the findings

All underlying data in the article have been provided.

Additional comments

To sum up, the content is of interest and innovation and this manuscript is recommended for publication.

Reviewer 3 ·

Basic reporting

The revised version of the manuscript seems improved.
The language is professional with clarity.
The background has been sufficiently described.
Article structure and tables are clear enough and the raw data look convincing.

I have a comment here in Figure 4 on how the heatmap was constructed. Can you please explain how can one plot the log2foldchange of expression values? As we would calculate the log2(fold change) with respect to control, one would be left with only values for PCOS.
And it is not possible to plot this graph with values for both the control and PCOS. Ideally, the values should be normalized and scaled for all the 8 samples and the heatmap should be constructed. Even the difference does not look convincing. The profiles of NC1 and NC2 do not look much different from PCOS samples.

Experimental design

The experimental design is well-defined and they have made the revisions as suggested improving the details of methods

Validity of the findings

The study is novel and there is no known study reporting the exosomal miRNAs in PCOS compared to matched controls. The rationale and benefit of the literature have been clearly stated.


The methodology and validity of Figure 4 needs to be checked.

Additional comments

The modifications are quite convincing except for Figure 4.

I have a minor comment.

The article has to be checked for grammatical errors.

---

## Round 0.3 · Minor Revisions

Please address the issues raised by Reviewer 3 concerning figures 2 and 4.

Reviewer 2 ·

Basic reporting

No comment.

Experimental design

No comment.

Validity of the findings

No comment.

Additional comments

No comment.

Reviewer 3 ·

Basic reporting

No new comments

Experimental design

No new comments

Validity of the findings

No new comments

Additional comments

1. The question of figure 2 is still not answered satisfactorily. Please explain how the heatmap was plotted. Were the expression values normalized and what method was used for clustering? Also please include a title for the legend (Expression or normalized expression).

2. The quality of Figure 4 (A and B) is poor. Please provide figures with better resolution.

---

## Round 0.4 · accepted · Accept

The authors have satisfactorily addressed the last points raised by Reviewer 3.

Reviewer 3 ·

Basic reporting

No additional comments

Experimental design

No additional comments

Validity of the findings

No additional comments

Additional comments

No additional comments